

# Edge mode locality in perturbed symmetry protected topological order

**Marcel Goihl***, **Christian Krumnow, Marek Gluza,**
**Jens Eisert and Nicolas Tarantino**

Dahlem Center for Complex Quantum Systems,
Freie Universität Berlin, 14195 Berlin, Germany

⋆ mgoihl@physik.fu-berlin.de

## Abstract

Spin chains with a symmetry-protected edge zero modes can be seen as prototypical systems for exploring topological signatures in quantum systems. However in an experimental realization of such a system, spurious interactions may cause the edge zero modes to delocalize. To combat this influence beyond simply increasing the bulk gap, it has been proposed to harness disorder which does not drive the system out of a topological phase. Equipped with numerical tools for constructing locally conserved operators that we introduce, we comprehensively explore the interplay of local interactions and disorder on localized edge modes in such systems. Contrary to established heuristic reasoning, we find that disorder has no effect on the edge mode localization length in the non-interacting regime. Moreover, disorder helps localize only a subset of edge modes in the truly interacting regime. We identify one edge mode operator that behaves as if subjected to a non-interacting perturbation, i.e., shows no disorder dependence. This implies that in finite systems, edge mode operators effectively delocalize at distinct interaction strengths despite the presence of disorder. In essence, our findings suggest that the ability to identify and control the best localized edge mode trumps any gains from introducing disorder.

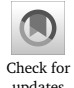

# 1   Introduction

Topological states of matter have been the focus of intense research over the past 30 years. Within systems of *condensed matter physics*, topological effects are known to occur in quantum Hall systems of the electron gas [1] and topological insulators [2]. Experiments on wires with proximity induced superconductors gave compelling evidence for Majorana zero modes [3]. Cold atomic gases and photonic devices offer possibilities of creating synthetic topological properties [4]. These new phases of matter by definition have no local order parameter but can be detected via their entanglement properties [5] and are classified by topological invariants [6]. When considering one-dimensional spin chains, these invariants give rise to protected gapless edge modes [7–14], which survive only if perturbations do not break the symmetries of the Hamiltonian.

Such edge modes are interesting from the perspective of quantum information science as well: They are one of many proposed candidates to encode quantum information robustly using topology [6,9,15]. However, the localization of the edge modes can be compromised by the onset of interactions allowing them to delocalize by hybridizing with delocalized bulk states. This will have deleterious effects on ones ability to encode and faithfully extract quantum information well before the topolgtical to trivial phase transition. As we can only expect to operate on a finite number of edge qubits to operate the quantum memory, the likelihood that a read-in/read-out procedure introduces errors increases with localization length, as more and more of the protected quantum information leaks into the bulk of the chain. To counteract this effect, it has been suggested that topological quantum information can be stabilized by disorder [16–18], which is supposed to inhibit transport by localizing the bulk. These works have given rise to the narrative that disorder is expected to always be beneficial when it comes to enhancing the localisation of the edge states.

The interplay of topological features, interactions and disorder is far from being fully understood. While there is evidence that disorder can drive a system into a topologically insulating phase [19–21], these do not in and of themselves support that any logical qubit is further localized by disorder. What is known rigorously is that, for topologically ordered systems, sufficiently weak local perturbations do not lift the ground-state degeneracy [22–24]

– but this kind of statement shows that small noise levels do not drive a phase transition, rather than making explicit constructive use of them. While this implicitly defines a coherence length for the edge modes, it is far from clear how the local structure of these operators is deformed in the presence of interactions and disorder. These seemingly basic questions should be addressed before more sophisticated scenarios can be meaningfully studied. This work sets out to do exactly that by studying the deformation of edge modes under disorder in a comprehensive fashion, laying the ground for a more general picture of the interplay of topological features and disorder.

In this work, we study the XZX cluster Hamiltonian, a topological chain which hosts one qubit at each edge and analyze if disorder can help localize them [17, 18] in the presence of weak interactions. We devise algorithms capable of calculating the support of the edge operators of the disordered XZX cluster Hamiltonian perturbed by either XX or XXZ type interactions. Equipped with this tool, we are in a position to determine the sensitivity of the edge mode localization length to each perturbation type. Contrary to previous expectations, we find that disorder only aids localization slightly, and only in the presence of interaction terms which are non-quadratic in the fermionic dual. Furthermore, and surprisingly, we also find that some edge modes are completely insensitive to disorder. Building on these findings, we elaborate on the lessons to be learned on the interplay of disorder and topological features.

## 2 SPT chains with spatial disorder

Our main focus is on the interplay of disorder, interactions and SPT order. Take for example a spin chain hosting a XZX cluster Hamiltonian

$$H_0(h) = -\sum_{j=2}^{N-1}(1 + h_j)X_{j-1}Z_jX_{j+1}, \tag{1}$$

where the $h_j$ are drawn uniformly from $\left[-\frac{\Delta}{2}, \frac{\Delta}{2}\right]$ and $X_j, Y_j, Z_j$ are the Pauli operators acting at site $j$. This system is known to be in a symmetry protected topological (SPT) phase, and thus supports localized, spin $1/2$, edge zero modes. The choice of disorder model here may seem unphysical to readers familiar with many-body localization, where a disordered local magnetic field is commonplace. Such a field competes with the SPT order, driving a transition to a topologically trivial phase. By disordering the cluster terms directly, we are implementing an ideal version of disorder, in that it breaks the degeneracies in the excited state spectrum while preserving the ground state manifold. Thus, if we should fail to observe increased localization in this circumstance we do not expect any improvement by moving to a more physical model of disorder.

Without any extra interaction terms, the two edge zero modes enforce a four-fold degeneracy at every energy level in the spectrum, and are perfectly localized on the two sites nearest to the boundary. By inspection, we can find local operators which exactly commute with the Hamiltonian (1)

$$\mathcal{E}_0 = \left\{ \begin{array}{ll} \mathcal{X}_L = X_1, & \mathcal{X}_R = X_L, \\ \mathcal{Y}_L = Y_1X_2, & \mathcal{Y}_R = X_{N-1}Y_N, \\ \mathcal{Z}_L = Z_1X_2, & \mathcal{Z}_R = X_{N-1}Z_N \end{array} \right\}, \tag{2}$$

that are located at the left and right edge. Apart from these local conserved quantities, the Hamiltonian in (1) also commutes with the time reversal operator

$$\mathcal{T} = \prod_{j=1}^{N} Z_j \mathcal{K}, \tag{3}$$

where $\mathcal{K}$ is complex conjugation. Note that all local edge operators fail to commute with time reversal and thus cannot be used to split the degeneracy without breaking the symmetry. Each set of these operators describe a spin-1/2 Hilbert space. Although $\mathcal{T}^2 = 1$ on the full Hilbert space, $\mathcal{T}^2 = -1$ when restricted to these spin-1/2 Hilbert spaces, i.e. these edge states transform projectively under the global time reversal (see Appendix A).

To perturb $H_0$, we will introduce a translationally invariant XXZ-coupling to our spin chain of the form

$$H_{\text{int}}(J, \eta) = -J \sum_{j=1}^{N-1} \left( X_j X_{j+1} + Y_j Y_{j+1} + \eta Z_j Z_{j+1} \right), \qquad (4)$$

which, for $J \ll 1$, is representative of interactions typically found in real solid-state material where Heisenberg-type interactions are ubiquitously present as a result of exchange interactions. Since we will only be investigating the effects of either an XX or a Heisenberg perturbation, we have included a parameter $\eta$ which interpolates between the two, and leave $J$ as the overall interaction strength. Specifically, we consider the following Hamiltonian defined on $N$ lattice sites

$$H(h, J, \eta) = H_0(h) + H_{\text{int}}(J, \eta), \qquad (5)$$

where we choose $\eta = 0$ or $\eta = 1$. By choosing $\Delta \neq 0$ we can switch on the presence of local disorder that can have the effect of diminishing the influence of the perturbation added to the exact Hamiltonian.

Note that $H_{\text{int}}(J, \eta)$ commutes with the time reversal operator for any values of $J$ and $\eta$, so if it is sufficiently weak it will only lift the degeneracy by an amount exponentially suppressed in system size [7–9, 25]. This occurs because, as soon as $J \neq 0$, the edge modes will no longer be perfectly localized at the edges and are in fact expected to be smeared within an exponential envelope [7–14]. With the degeneracy lifted, the edge mode operators will no longer commute exactly with the Hamiltonian, since the existence of operators which anticommute with $\mathcal{T}$ (a feature of the edge modes in (2)) and commute with Hamiltonian would require *exact* degeneracies due to Kramer's theorem. The failure of the edge mode operators to commute exactly with play an important role in informing our algorithm in Section 3.2.

From here, we set out to understand this interplay of topology, interactions and disorder by explicitly constructing edge modes for this perturbed XZX cluster Hamiltonian. For Anderson and many-body localized systems, the localization length of all local conserved quantities depends on the disorder strength. Thus, one would expect that localizing the bulk of the SPT chain should stabilize the edge modes as excitations cannot traverse the full system to allow the hybridization of opposite edges [17, 18]. We devise methods capable of computing the edge mode support in presence of both non-interacting and many-body interacting perturbations. Contrary to the previously stated heuristic argument, we find numerous cases where the edge modes are completely insensitive to disorder.

## 3 Edge mode construction

In this section, we describe two methods employed to construct the edge mode operators $\mathcal{E}$ in the perturbed XZX cluster Hamiltonian. Computing the broadening of the edge modes is a particularly daunting task, precisely because the operator encodes information about states throughout the entire spectrum, and thus cannot be studied using low energy techniques such as DMRG. The first method assumes that the perturbed Hamiltonian represents non-interacting fermions and yields an efficient solution in terms of Majorana eigenmodes, which

allows for a direct computation of the edge modes. The second approach tackles the generic interacting case, where no Bogoliubov transformation will suffice and hence the construction of the edge modes becomes more intricate. In this case we rely on a method developed to construct conserved operators called "l-bits" for a many-body localized system [26, 27]. The construction of these conserved quantities from first principles is difficult, but algorithms which can construct them using various methods do exist [28–40].

## 3.1 Edge modes under free fermion perturbations

After a Jordan-Wigner transformation, the choice of $\eta = 0$ is equivalent to a non-interacting fermionic problem whereas $\eta \neq 0$ maps to an interacting fermionic model with quartic interactions. Introducing the Majorana operators $\bar{\gamma}_j, \gamma_j$ for $j = 1, \dots, N$ as

$$\gamma_j = Z_1 \dots Z_{j-1} X_j \quad \text{and} \quad \bar{\gamma}_j = Z_1 \dots Z_{j-1} Y_j, \tag{6}$$

with

$$\{\gamma_j, \gamma_k\} = \{\bar{\gamma}_j, \bar{\gamma}_k\} = 2\delta_{j,k}, \quad \{\bar{\gamma}_j, \gamma_k\} = 0, \tag{7}$$

the Hamiltonian (5) becomes

$$H(h, J, \eta) = -i \sum_{j=2}^{N-1} (1 + h_j) \bar{\gamma}_{j-1} \gamma_{j+1} - J \sum_{j=1}^{N-1} \left( i\bar{\gamma}_j \gamma_{j+1} + i\gamma_j \bar{\gamma}_{j+1} - \eta \gamma_j \bar{\gamma}_j \gamma_{j+1} \bar{\gamma}_{j+1} \right), \tag{8}$$

which is non-interacting if and only if $\eta = 0$. Written in terms of the Majorana operators, the edge modes for $J = 0$ take the form

$$\begin{aligned} \mathcal{X}_L &= \gamma_1, \quad \mathcal{Y}_L = i\gamma_1\gamma_2, \quad \mathcal{Z}_L = \gamma_2, \\ \mathcal{X}_R &= -iP\bar{\gamma}_N, \quad \mathcal{Y}_R = -i\bar{\gamma}_{N-1}\bar{\gamma}_N, \quad \mathcal{Z}_R = -iP\bar{\gamma}_{N-1}, \end{aligned} \tag{9}$$

with $P = Z_1 \cdots Z_N$ being the global parity operator which commutes with $H$. For this we note that (8) can be written as

$$H(h, J, \eta = 0) = i \sum_{j,k=1}^{N} \gamma_j C_{j,k} \bar{\gamma}_k, \tag{10}$$

with the coupling matrix

$$C_{i,j} = \begin{cases} J & \text{if} \quad i = j+1 \\ -J & \text{if} \quad i = j-1 \\ -(1 + h_{i+1}) & \text{if} \quad i = j-2. \end{cases} \tag{11}$$

As $C \in \mathbb{R}^{N \times N}$ is real, the singular value decomposition of $C$ takes the specialized form $C = Q^T \Sigma \bar{Q}$ with two orthogonal $Q, \bar{Q} \in O(N)$ and $\Sigma \in \mathbb{R}^{N \times N}$ a diagonal matrix with real non-negative entries. Using the two orthogonal matrices $Q, \bar{Q} \in O(N)$ we introduce new modes

$$m_j = \sum_{k=1}^{n} Q_{j,k} \gamma_k, \qquad \bar{m}_j = \sum_{k=1}^{N} \bar{Q}_{j,k} \bar{\gamma}_k, \tag{12}$$

which again fulfill the Majorana anti-commutation relations (7) and the Hamiltonian (10) becomes diagonal taking the form

$$H(h, J, \eta = 0) = i \sum_{l=1}^{N} \sigma_j m_j \bar{m}_j, \tag{13}$$

where we have defined the single particle energies $\sigma_j = \Sigma_{j,j}$ and assume without loss of generality that they are in increasing order.

For $J = 0$, we find that $\sigma_1 = \sigma_2 = 0$ with $m_1 = \gamma_1$, $m_2 = \gamma_2$, $\bar{m}_1 = \bar{\gamma}_n$, $\bar{m}_2 = \bar{\gamma}_{n-1}$ being the corresponding localized edge mode operators. At finite $J > 0$, $\sigma_1 \sim \sigma_2 \sim e^{-n/n_0}$ are not exactly zero anymore but decay exponentially with increasing system size [9] and hence much smaller compared to the next largest value $\sigma_3$. Thus an approximate four-fold degenerate ground-state sector remains well defined. The operators $m_1, m_2, \bar{m}_1, \bar{m}_2$ hence correspond to perturbed edge mode operators which can be individually studied in the free fermionic setting via their single particle wavefunctions. Note however, that only their products $m_1 \bar{m}_1$ and $m_2 \bar{m}_2$, which are supported at both ends of the chain, are exact constants of motions of the Hamiltonian. This will also be a main difference to the interacting relaxation algorithm which from the outset seeks operators that exactly commute with the Hamiltonian.

## 3.2 Edge modes under perturbative many-body interactions

The intuition behind our approach to constructing edge modes of a system with many-body interactions is as follows: the edge modes $\mathcal{E}_0$ of the unperturbed model $H_0$ should deform smoothly to those of the full interacting Hamiltonian. Indeed, they turn out to be good starting points to obtain the actual edge mode operator $\mathcal{E}$ which commutes with $H$ exponentially well in the system size whilst remaining local to some degree.

The method requires an ansatz which is expected to resemble the conserved operators. Since we are perturbing away from a solvable point, we employ a natural choice, the exact edge modes obtained in the unperturbed fixed-point model. One might be inclined to use a single edge mode as an ansatz for finding the perturbed operators, but this approach will fail in general as the operators produced with this method necessarily commute exactly with the Hamiltonian by construction. This is not the case for single edge modes as they *cannot* commute with Hamiltonian, as discussed in Section 2. We can circumvent this problem by instead using products of edge modes supported on both the left and right ends of the chain which we call $\mathcal{B}_0 = \mathcal{E}_0^L \otimes \mathcal{E}_0^R$. Such products respect the time reversal symmetry and thus are not prevented from commuting exactly with the Hamiltonian. Due to the topological degeneracies present in our model, we have to make sure that the basis in any subspace also diagonalizes our edge mode guess $\mathcal{E}_0$ if we want to obtain the form in Eq. (15). This is reminiscent of standard degenerate perturbation theory and in fact requires by far the most ressources of the total algorithm as we need to rediagonalize $2^{N-2}$ many $4 \times 4$ matrices.

We will now detail how to construct the quasi-local conserved operators. By definition, these have a compact representation in the energy basis. This basis, which we label by $\{|k\rangle\}$, is obtained from full exact diagonalization. Consider a basis for diagonal operators in energy space fulfilling the Pauli-algebra. The minimal elements of this basis may take the following form

$$\Xi_i := \mathbb{1}_{2^{i-1}} \otimes Z \otimes \mathbb{1}_{2^{N-i}} = \sum_{k=1}^{2^N} (-1)^{\lfloor (k-1)/2^{N-i} \rfloor} |k\rangle\langle k| , \tag{14}$$

where $Z$ is a Pauli-operator. This might look complicated at first glance, but it is really nothing but Pauli-operators defined in energy space. The full basis can be obtained by calculating all products of these $N$-many operators. These operators by construction exactly commute with the Hamiltonian. Hence, any constant of motion can be brought into the form of the $\Xi$ operators. Their real space representation $U_D \Xi_i U_D^\dagger$ will however in general not be local. Their locality is completely dependent on the ordering of the eigenstates in the unitary $U_D$ as this is the only freedom left. Due to the immense number of possible permutations – the order of the symmetric group $S_{2^N}$ is $2^N!$ – a brute force approach is out of scope. We instead rely on a heuristic algorithm which dynamically relaxes an ansatz operator to obtain a good

permutation. Abstractly speaking, we bank on the time independent or equilibrium part of $\mathcal{B}_0$ to resemble the product of the perturbed edge modes $\mathcal{B}$ already quite well. In cases where this is not given, the algorithm will fail to produce a local edge mode.

The unperturbed operator $\mathcal{B}_0$ and a diagonalization unitary $U_D$ serve as inputs to our method. This unitary has an arbitrary ordering of eigenvectors at the start (for ED, usually determined by the size of the energies of the Hamiltonian). Upon mapping $\mathcal{B}_0$ to its equilibrium representation, we use it to obtain an ordering of the eigenstates that resembles the Pauli structure well. This representation is obtained by calculating the *infinite time average* of $\mathcal{B}_0$, which stems from *equilibration theory*

$$\mathbb{E}(\mathcal{B}_0) := \lim_{T \to \infty} \frac{1}{T} \int_0^T dt \, \mathcal{B}_0(t) = \sum_k \langle k | \mathcal{B}_0 | k \rangle | k \rangle \langle k | , \tag{15}$$

the time average will hence be diagonal in the energy eigenbasis for non-degenerate spectra and is thus a constant of motion. However, since the infinite time average is not trace preserving, it in general causes $\mathcal{B}_0$ to lose its algebraic structure. We would like to point out that while localizing systems are in general not expected to thermalize, they do equilibrate which makes this ansatz meaningful [41].

Due to the topological degeneracies present in our model, we have to make sure that the basis in any subspace also diagonalizes our edge mode guess $\mathcal{E}_0$ if we want to obtain the form in Eq. (15). This is reminiscent of standard degenerate perturbation theory and in fact requires by far the most resources of the total algorithm as we need to rediagonalize $2^{N-2}$ many $4 \times 4$ matrices.

We then set out to find a permutation of the eigenvectors of $H$ such that the time average of the $\mathcal{B}_0$ best resembles $\Xi_1$ which can be done by a sorting of the eigenvalues of $\mathbb{E}(\mathcal{B}_0)$. This permutation $\mathcal{P}$ also gives rise to a new diagonalization unitary $\widetilde{U_D} = \mathcal{P} U_D$. Upon conjugating $\Xi_1$ with $\widetilde{U_D}$, we obtain an edge mode which fulfills all algebraic properties and commutes with the Hamiltonian. Since our sorting method is heuristic, we cannot rule out the existence of better localized edge modes. Nevertheless, the support that we find serves as a robust upper bound. Because of this, we note that a breakdown of our method, i.e. finding a non-local operator, does not necessarily imply that there are no localized edge modes.

The following pseudocode describes a possible way to implement this procedure numerically. We use a notation close to python.

```
1  input: diagonalizing unitary U_D (as obtained from ED and
2  rediagonalization in degenerate subspaces)
3  input: edge modes of the unperturbed system E_0^L, E_0^R
4  output: quasi-local diagonalization unitary U_D
5
6  define infinite_time_average(V, O):
7      return diag(VOV†)
8
9  spec = infinite_time_average(U_D, E_0^L ⊗ E_0^R)
10 perm = argsort(spec)
11
12 return U_D[:,perm]
```

This algorithm builds on a previously introduced method used in the context of many-body localized systems [40]. In this problem, the authors designed operators which commute exactly with the given Hamiltonian and are quasi-local. In a *many-body localized system*, one searches for extensively many quasi-local constants of motion and the system features a fully non-degenerate spectrum caused by the disordered potential landscape. In contrast, the SPT model is characterized by only constantly many edge mode operators which enforce degeneracies throughout the spectrum. These differences necessitated major modifications to the method from Ref. [40].

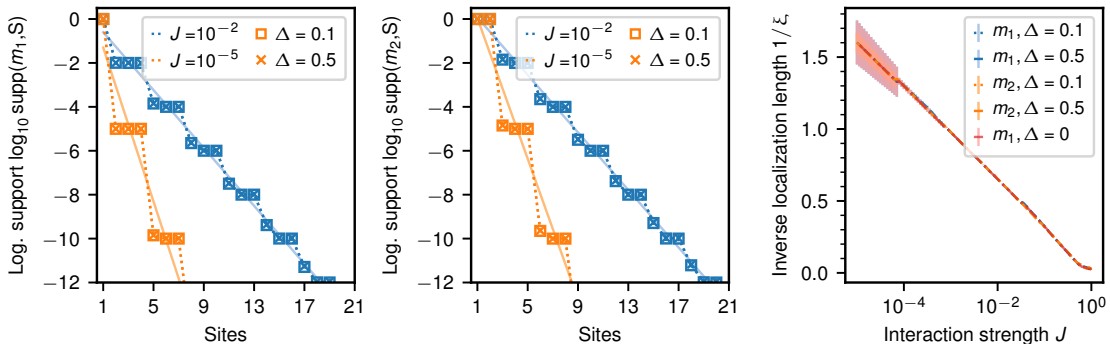

Figure 1: Left, Center: Logarithmic support $\log_{10}\mathrm{supp}(\mathcal{B},S)$ of the edge modes $m_1$ and $m_2$ for $\eta = 0$ on $N = 32$ sites, where $S$ is a region starting from the left end of the system where sites starting from the right end have been removed. Color encodes the interaction (or hopping) strength $J$ and markers indicate the disorder strength $\Delta$. Each data point is an average over 100 realizations with error bars indicating the standard deviation of the data (smaller than symbols). Dotted lines are guides to the eye. Solid lines show linear fits of the data for $\Delta = 0.1$ , which allow to extract the localization length $\xi$. Right: Interaction dependence of the inverse localization length $1/\xi$ for the edge modes. Here, color encodes the two modes and disorder strength $\Delta$. Data for all interaction strengths and modes overlaps strongly. The lines shown have been extrapolated by using 100 values of interaction strengths in the shown interval. We also include data without disorder. Here, the errorbars show the quality of the fit in form of the least-squares error.

## 3.3 Measure of locality

In the following analysis we set out to assess the locality of the constructed edge mode operators. Therefore, we want to compare the action of the full operator to a itself truncated to a local region only. As a first step, we will need to specify a reduction map, which reduces our operators to such operators with local support in a region $S$

$$\Gamma_S(A) := \frac{1}{2^{S^C}} \, \mathrm{tr}_{S^C}(A) \otimes 1_{S^C} \,, \tag{16}$$

where support is defined using site indices. This map truncates an operator down to its local support on $S$ and afterwards embeds it into the full real space again by tensoring identities on $S^C$. This operator can now be compared to the original operator supported on the full system. The difference between the two will be a measure of the support

$$\mathrm{supp}(A,S) = \|A - \Gamma_S(A)\|_\infty \,. \tag{17}$$

Due to the interacting procedure yielding products of edge modes, we expect their support to be mainly on both edges of the system. To assess their locality, we hence use an $S$ which is centered in the middle of the chain and extends by increasing this block on its both ends by one site. We note that the norm used here is most sensitiveand in many other applications operators which are expected to be local are so only in weaker norms than in operator norm [39, 40, 42, 43].

In the non-interacting case $\eta = 0$ we have access to the individual edge mode operators $m_1$, $m_2$, $\bar{m}_1$, $\bar{m}_2$ and we hence consider the sets $S_{L,k} = [k]$ and $S_{R,k} = [N-k]^C$ oriented at the left and right boundary of the system. The larger system size considered in this case, prohibits to use the full Hilbert-space representation of the operators. However, as we show in the appendix, one can exploit the algebraic properties of the non-interacting fermions in order

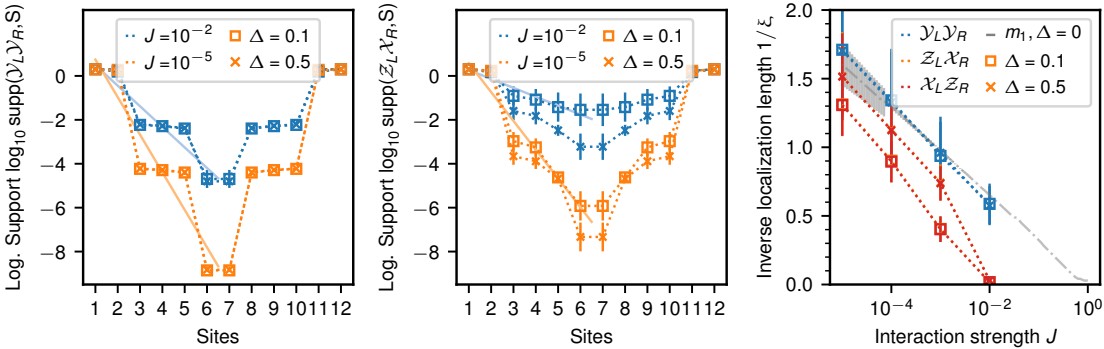

Figure 2: (Left, center) Logarithmic support $\log_{10}\mathrm{supp}(\mathcal{B}, S)$ of the edge modes $\mathcal{Y}_L \mathcal{Y}_R$ and $\mathcal{Z}_L \mathcal{X}_R$ for $\eta = 1$ on $N = 12$ sites, where $S$ is the left and right part of the system where blocks of even size centred around the middle of the chain have been removed. Colour encodes the used interaction strength $J$ and markers encode the disorder strength $\Delta$. Each data point is an average over 100 realizations with error bars indicating the standard deviation of the data. Dotted lines are a guide to the eye. Solid lines show linear fits of the data for $\Delta = 0.1$, which allow to extract the localization length $\xi$. Right: Interaction dependence of the localization length $\xi$ for all three edge modes. Here, color encodes the three modes and markers again encode disorder strength $\Delta$. The data for $\mathcal{Z}_L \mathcal{X}_R$ and $\mathcal{Z}_R \mathcal{X}_L$ overlaps completely which is why it is hard to spot the orange markers indicated in the legend. Again, the errorbars show the least-squares errors of the fit. The grey line is taken from the non-interacting results as a comparison.

to directly compute the reduction and norm of it within the fermionic picture which yields for $p = 1, 2$

$$\| m_p - \Gamma_{S_{L,k}}(m_p) \| = \sqrt{\sum_{l=k+1}^{N} Q_{l,p}^2}, \tag{18}$$

$$\| P\bar{m}_p - \Gamma_{S_{R,k}}(P\bar{m}_p) \| = \sqrt{\sum_{l=1}^{N-k} \bar{Q}_{l,p}^2}. \tag{19}$$

## 4 Numerical Results

In this section, we show and discuss the resulting operators for both models. We have worked with at least four interaction strengths $J \in \{10^{-2}, 10^{-3}, 10^{-4}, 10^{-5}\}$ and three disorder strengths $\Delta \in \{0.1, 0.3, 0.5\}$. For a clearer presentation, we only picked a subset of these results but the calculations not shown behave analogously.

### 4.1 Free fermionic perturbation

For $\eta = 0$, as described above, we show the support of the single edge modes supported on the left part of the chain. The system has a total size of $N = 32$ sites. While much larger systems are treatable and have been investigated by us with this algorithm, we find that this system size suffices to properly display the edge mode decay. For a system size scaling of this method, we refer interested reader to the Appendix D. Results can be found in Fig. 1. Left

and center plots show the logarithmic support $\log_{10}\text{supp}(\mathcal{E}, S)$ of the edge modes $m_1, m_2$. We use the data of these plots to extract a localization length $\xi$, shown in the right plot. The errorbars indicating the least-sqares error of the fit for small interactions stem from the fact that in these systems, the support of the edge mode falls of very strongly yielding only few non-zero points and therefore less accurate fits. We find that the inverse localization length $\xi$ depends logarithmically on the interaction strength $J$.

The different modes $m_1$ and $m_2$ show the same qualitative behaviour. The support falls off in exponential fashion with the size of the support region. This aligns nicely with the intuition that additional interaction terms should only dress the original modes. Furthermore, the observed plateaus can be derived for the infinite system size limit as shown in Appendix C. With increasing interaction strength $J$, the edge modes become less local, as expected from perturbation theory.

A feature of particular note in these results is the insensitivity of the edge mode locality to the disorder strength $\Delta$. As a comparison, we also show data without any disorder. This is surprising when contrasted with the intuition that disorder should help localize the edge modes [17, 18]. This suggests that the edge modes of this SPT do not couple to the bulk operators in circumstances of Anderson localization.

## 4.2 Many-body interacting perturbation

Now we resort to the calculations performed for $\eta = 1$, corresponding to the interacting system. Due to the size of the Hilbert space and the effort of the re-diagonalization of the topologically degenerate subspaces, we had to resort to system size $N = 12$. Fig. 2 again shows the logarithmic support $\log_{10}\text{supp}(\mathcal{E}, S)$ for $\mathcal{E} \in \{\mathcal{Y}_L \mathcal{Y}_R, \mathcal{X}_L \mathcal{Z}_R\}$ on the left and center panel. The right panel shows the extracted localization length. A more detailed discussion on the fitting procedure and cross validation of the code can be found in Appendix D. The symmetry of the plot is due to the choice of the system $S$ as laid out in section 3.3.

The support again shows an exponential decay of operator support into the bulk. Moreover, this localization length $\xi$ grows with increasing the interaction strength $J$ as expected. The localization length observed for all edge modes is of the same order as the one found in much larger system sizes for the non-interacting perturbation (cp. gray dash-dotted line). However, when increasing the interaction strength to $J = 10^{-2}$ there is a sharp drop in the localization length for some operators which might be ascribed to a transition towards a topologically trivial phase. This transition point is far lower than the expected value of $J \sim 1$. This is an expected finite size effect as the edge modes are a priori closer together and thus able to hybridize more easily. Furthermore, the fit errors shown in this plot stem can also be ascribed to the finite size of the interacting system, since we are effectively fitting very few points. Nevertheless, the fit errors allow for distinguishing the different behavior of the modes. Nevertheless, the compatibility between non-interacting and many-body interacting localization lengths away from this transition indicates that the signal of SPT behaviour can still be reliably observed in system sizes tractable by exact diagonalization.

For the $\mathcal{Z}_L \mathcal{X}_R$ mode we find that the heuristic picture is recovered as increasing disorder strength aids localization. This contrasts strongly with our findings in the non-interacting case, indicating that many-body interactions are necessary to couple the edge modes to the bulk operators. Moreover, the localization length is generically longer than in the free fermion case with disorder strength pushing the length down towards the free fermion value. This suggests that the free fermion value represents the best localization of the edge modes for fixed interaction strength. The same behaviour is found for $\mathcal{X}_L \mathcal{Z}_R$ (cp. see appendix).

An exception to the behaviour reported above is displayed in the localization length of the $\mathcal{Y}_L \mathcal{Y}_R$ edge mode. Despite the presence of many-body interactions, it shows a disorder insensitivity akin to that of the non-interacting regime. This goes beyond mere analogy as

the value of the localization length of the $\mathcal{Y}_L \mathcal{Y}_R$ operator overlaps perfectly with the non-interacting results. We computed the localization behaviour for all six possible edge mode hybridization patterns and found that only the $\mathcal{Y}_L \mathcal{Y}_R$ operator displays free fermion localization behaviour. This suggests that this mode is subject to a selection rule which precludes the many-body interaction effects which delocalize the $\mathcal{Z}_L \mathcal{X}_R$ mode. The source of this selection rule is at this point mysterious, but we note that the $\mathcal{Y}_L \mathcal{Y}_R$ operator is unique among the choices of edge mode products in being local to the edges in both spin and fermionic variables, i.e. it does not feature a parity string across the whole chain. The absence of such a non-local feature in the fermionic picture may explain the reduced sensitivity to bulk localization behaviour.

Put succinctly, our results suggest that in the presence of many-body interactions, there may be a splitting of the modes into those which delocalize faster, i.e. $\mathcal{Z}_L \mathcal{X}_R$ and $\mathcal{X}_L \mathcal{Z}_R$ and are sensitive to disorder and one mode $\mathcal{Y}_L \mathcal{Y}_R$, which is insensitive to disorder and shows a stronger localization comparable to the one of non-interacting edge modes. We would like to point out, that since our method can only provide upper bounds to the localization behaviour, it is still conceivable that all three modes behave the same. Also, it is possible that the disorder sensitivity observed in all other products vanishes in larger systems than we are able to treat. However even if a finite size effect, this splitting constitutes an interesting result as it would be relevant for short synthetic chains or cold ion systems. In such circumstances where one seeks to improve edge mode locality in presence of many-body interactions to encode quantum information, the gains from disorder potentials are marginal compared to those from picking "better" edge modes.

## 5 Conclusions

In this work, we investigated the localization behavior of topological edge mode operators upon introducing both non-interacting and many-body interacting perturbations as well as disorder. Specifically, we started out from the disordered XZX cluster Hamiltonian which as a fixed-point model is exactly soluble and added XX and XXZ interactions which are expected to drive the transition towards a topologically trivial model. We introduce different methods of finding the topological edge mode operators, one based on the Majorana description which yields the lowest lying eigenmodes for non-interacting systems and a second one, which uses the relaxation of the fixed-point edge modes as an ansatz to heuristically find local edge modes for many-body interacting chains. While the support of the obtained edge operators with the interacting method is only an upper bound, the commutation with the Hamiltonian is exact.

Both perturbations considered delocalize as their strength is increased. However, the non-interacting model displays no disorder dependence whereas the interacting system does. Curiously, a single edge mode combination which in the fermionic language corresponds to the two density operators at both ends, namely $\mathcal{Y}_L \mathcal{Y}_R$, shows no disorder dependence even when adding many-body interactions. Our results suggest that for a finite size chain, one might find different localization behavior for different edge mode operators. Specifically, we find one edge mode that is most stable and completely insensitive to disorder, picking it out as the one best-suited to encode a logical qubit.

Since we fully diagonalize the Hamiltonian we are limited to small system sizes even for this one-dimensional problem. We hope to extend the method to larger systems by truncating to the ground state sector, which would possibly allow a tensor network implementation as well. The interacting method used in this work relies only on guessing a suitable ansatz edge mode operator. Hence, we plan to apply it to more physical models and other types of perturbations such as open dynamics. Here, one might hope to overcome the thermal instability of topological systems [44] with the help of disorder [45].

## 6  Acknowledgements

This work has been supported by the ERC (TAQ), the DFG (CRC 183, FOR 2724, EI 519/7-1), and the Templeton Foundation. This work has also received funding from the European Union's Horizon 2020 research and innovation programme under grant agreement No 817482 (PASQuanS).

## A  Fractionalization

To see this, we must rewrite the time-reversal operator in a way that makes the edge action explicit. We can do this by re-expressing our time reversal operator using the cluster operators of our Hamiltonian,

$$\prod_{j=2}^{N-1} X_{j-1} Z_j X_{j+1} = (-1)^N X_1 X_2 \left( \prod_{j=2}^{N-1} Z_j \right) X_{N-1} X_N \tag{20}$$

$$= (-1)^N Y_1 X_2 \left( \prod_{j=1}^{N} Z_j \right) X_{N-1} Y_N \tag{21}$$

$$= \mathcal{Y}_L \left( (-1)^N \prod_{j=1}^{N} Z_j \right) \mathcal{Y}_R, \tag{22}$$

which lets us recast $\mathcal{T}$ as

$$\mathcal{T} = \mathcal{Y}_L \left( \prod_{j=2}^{N-1} X_{j-1} Z_j X_{j+1} \right) \mathcal{Y}_R \mathcal{K}. \tag{23}$$

If we decompose our Hilbert space into edge and bulk tensor factors, we can identify the emergent edge action

$$\mathcal{T} = \mathcal{Y}_L \mathcal{K}_L \otimes \left( \prod_{j=2}^{N-1} X_{j-1} Z_j X_{j+1} \mathcal{K}_{bulk} \right) \otimes \mathcal{Y}_R \mathcal{K}_R \tag{24}$$

and so we can see that the localized operators of time reversal on the edge states are

$$\mathcal{T}_{L/R} = \mathcal{Y}_{L/R} K_{L/R} \tag{25}$$

which, curiously, these operators do not square to 1. Instead,

$$\mathcal{T}_{L/R}^2 = \mathcal{Y}_{L/R} K \mathcal{Y}_{L/R} K_{L/R}, \tag{26}$$

$$= \mathcal{Y}_{L/R} (-\mathcal{Y}_{L/R}) K_{L/R}^2 = -1, \tag{27}$$

which demonstrates the symmetry fractionalization expected in an SPT phase.

## B  Operator norm decay of edge modes for $\eta = 0$

In the following we explicitly compute the norm of the edge mode operators $m_p$ and $P\bar{m}_p$ with $p = 1, 2$ obtained in the case $\eta = 0$. Here, it is important to keep in mind, that we want to study

the effect of the reduction map on the level of the qubits in which the original Hamiltonian in (1) is defined. All following norms and traces are hence evaluated by reversing the Jordan-Wigner transformation and relating the fermionic operators to the Pauli operators. The results are then mapped again to the fermionic level as the expressions take a more compact form here.

Due to the non-interacting structure of the problem, the edge mode operators are linear combinations of the initial Majorana operators $\gamma_j$, $\bar{\gamma}_j$. In this case the result of the reduction map can be studied in detail. We find

$$
\begin{aligned}
\mathrm{tr}_{S_{L,k}^C}(m_p) &= \sum_{l=1}^{k} Q_{p,l}\gamma_l, \\
\mathrm{tr}_{S_{R,k}^C}(P\bar{m}_p) &= \sum_{l=N-k+1}^{N} \overline{Q}_{p,l} P\bar{\gamma}_l,
\end{aligned}
\tag{28}
$$

where, as explained above, the trace is evaluated on the level of the qubits. The differences $A - \Gamma(A)$ are then given by

$$
\begin{aligned}
m_p - \Gamma_{S_{L,k}}(m_p) &= \sum_{l=k+1}^{N} Q_{p,l}\gamma_l, \\
P\bar{m}_p - \Gamma_{S_{R,k}}(P\bar{m}_p) &= \sum_{l=1}^{N-k} \overline{Q}_{p,l} P\bar{\gamma}_l,
\end{aligned}
\tag{29}
$$

and again essentially only linear combinations of $\gamma_j$ and $\bar{\gamma}_j$.

We can however compute easily the norm of any linear combination of Majorana operators. Let $S \subset [N]$ and define

$$
A = \sum_{j\in S} a_j \gamma_j
\tag{30}
$$

to be any linear combination of the $\gamma_j$ operators with $a_j \in \mathbb{R}$ for $j \in S$. One finds that the square of $A$ is given by

$$
\begin{aligned}
A^2 &= \sum_{j,k\in S} a_j a_k \gamma_j \gamma_k \\
&= \sum_{j\in S} a_j a_j \mathbb{1} + \sum_{j,k\in S:j<k} a_j a_k \gamma_j \gamma_k + \sum_{j,k\in S:k<j} a_j a_k \gamma_j \gamma_k \\
&= \sum_{j\in S} a_j^2 \, \mathbb{1}.
\end{aligned}
\tag{31}
$$

From this we can directly conclude, that $A$ has only two degenerate eigenvalues $\pm(\sum_{j\in S} a_j^2)^{1/2}$ such that

$$
\|A\|^2 = \sum_{j\in S} a_j^2
\tag{32}
$$

can be directly computed. The same argument holds for the operators $P\gamma_j$. Hence, we obtain

$$
\begin{aligned}
\|m_p - \Gamma_{S_{L,k}}(m_p)\|^2 &= \sum_{l=k+1}^{N} Q_{p,l}^2, \\
\|P\bar{m}_p - \Gamma_{S_{R,k}}(P\bar{m}_p)\|^2 &= \sum_{l=1}^{N-k} \overline{Q}_{p,l}^2.
\end{aligned}
\tag{33}
$$

## C  Scaling behaviour of support results for $\eta = 0$

Some features of the edge mode support can be inferred from analytical results computed for $N \to \infty$. Starting with

$$H = -i \sum_{j=2}^{N-1} \Delta_j \bar{\gamma}_{j-1} \gamma_{j+1} - J \sum_{j=1}^{N-1} \left( i\bar{\gamma}_j \gamma_{j+1} + i\gamma_j \bar{\gamma}_{j+1} \right), \tag{34}$$

we can attempt to construct the edge modes iteratively. We can infer from the structure of the Hamiltonian that the left edge modes will be of the form

$$m_p = \sum_{j=1}^{N} \alpha_{p,j} \gamma_j. \tag{35}$$

In the thermodynamic limit, the edge zero modes are expected to be exact. This imposes a condition on the coefficients of the zero modes via the commutator

$$\frac{1}{2i} \left[ m_p, H \right] = \bar{\gamma}_1 \left( J\alpha_{p,2} + \Delta_2 \alpha_{p,3} \right) + \sum_{j=2}^{\infty} \bar{\gamma}_j \left( J\alpha_{p,j+1} + \Delta_{j+1} \alpha_{p,j+2} - J\alpha_{p,j-1} \right) = 0, \tag{36}$$

from which we can construct a recurrence relation

$$\alpha_{p,j+1} = -\frac{J}{\Delta_j} \left( \alpha_{p,j} - \alpha_{p,j-2} \right), \tag{37}$$

that willll allow us to produce two linearly independant edge mode operators. Since we assume that these should be smoothly related to the perfectly localised operators when $J = 0$, we choose to begin the recurrance relation with either $\alpha_1 = 1$ and $\alpha_2 = 0$, which we identify with $m_1$, or vice versa, which we identify with $m_2$. In the case of $m_1$, a clear scaling behaviour emerges from $j = 4$ onward

$$\alpha_{1,3k+1} \sim J^{-k}, \quad \alpha_{1,3k+2} \sim J^{-(k+1)}, \quad \alpha_{1,3k+3} \sim J^{-(k+2)}, \quad k \in \mathbb{N}, \tag{38}$$

while the coefficients of $m_2$ exhibit a similar scaling behaviour from $j = 3$ onward

$$\alpha_{2,3k} \sim J^{-k}, \quad \alpha_{2,3k+1} \sim J^{-(k+1)}, \quad \alpha_{2,3k+2} \sim J^{-(k+2)}, \quad k \in \mathbb{N}, \tag{39}$$

which predicts a significant amount of structure in our support measure plots. Since we can compute our support measure exactly in the free fermion context, we see that

$$\mathrm{supp}\left( m_1, S_{L,j} \right) \sim J^{-\left\lfloor \frac{j-1}{3} \right\rfloor}, \tag{40}$$

$$\mathrm{supp}\left( m_2, S_{L,j} \right) \sim J^{-\left\lfloor \frac{j}{3} \right\rfloor}, \tag{41}$$

from which we infer that the support measure should have decending plateaus of width 3, all of which fall within an exponential envelope, which is precisely what is observed in Figure 1. The analysis for the right edge modes is identical, and we expect these to also show a plateau structure, which is also seen.

## D  Finite size scaling and cross validation

In this appendix, we discuss the finite size scaling of the non-interacting code and also explain the cross validation between the two methods.

The data for the finite size scaling for the non-interacting ($\eta = 0$) code is shown in Fig. 3. The color encodes support data of $m_0$ for different different system sizes and the gray line is machine precision. Each point is an average over 100 realizations for interaction and disorder values similar to the main text. We find that the different system sizes agree quite well in the parameter regime investigated here. Moreover, the data indicates that at a size of 24 sites the support decays to the machine precision, which is why we settled for a system size of that order for the main text material despite the availability of even larger systems.

For the interacting code things are quite different, as here 12 sites is the maximal system size that can be reached due to the effort needed for the sorting procedure. Furthermore, the emergent plateau structure which can be demonstrated in the infinite system limit for the non-interacting case (see App.,C) still persists in the interacting model. Thus, we are forced to effectively fit the exponential envelope to only three points, which unfortunately renders a system size scaling towards smaller systems meaningless. Nevertheless, the interacting procedure naturally also works if the Hamiltonian is actually non-interacting so we used this to at least cross validate results between both algorithms for small systems where they agree.

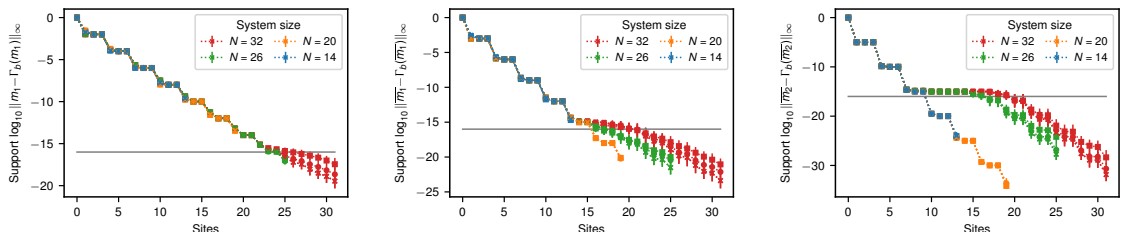

Figure 3: Finite size scaling for the non-interacting ($\eta = 0$) algorithm and the $m_0$ mode. Color encodes system size, the gray line indicates machine precision. Each point is an average over 100 realizations. The different panels show interaction strength $J = 10^{-2}, 10^{-3}, 10^{-5}$ from left to right. The three values of disorder $\Delta = 0.1, 0.3, 0.5$ are indicated by markers and either lie on top of one another or are below machine precision.

## E  Additional numerical data

In this appendix, we show additional numerical data obtained for the $\mathcal{X}_L \mathcal{Z}_R$ edge mode and a bulk operator. Fig. 4a shows the support of $\mathcal{X}_L \mathcal{Z}_R$ edge mode on the same scale as in the main text. It shows the same disorder dependence and has the same localization length as $\mathcal{Z}_L \mathcal{X}_R$. Fig. 4b shows the localization behaviour of a bulk operator in a chain of length $N = 11$. Here again, the disorder strength decreases the localization length.

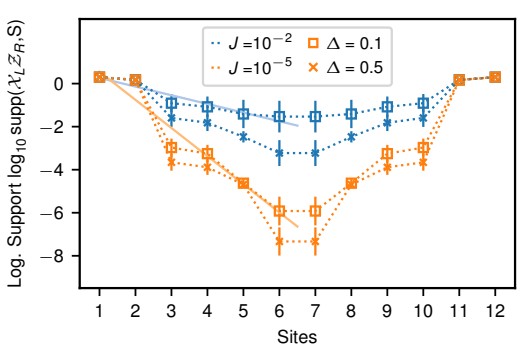
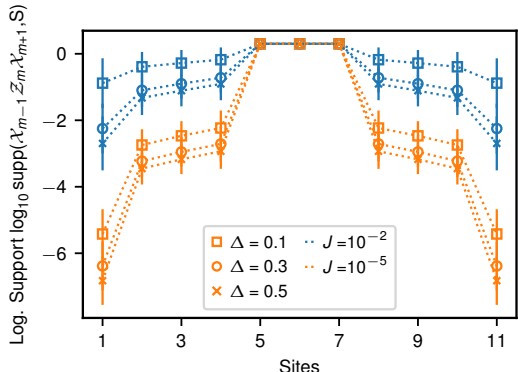

(a) Logarithmic support $\log_{10}\mathrm{supp}(\mathcal{X}_L\mathcal{Z}_R, S)$ for $\eta = 1$ on $N = 12$ sites. , where $S$ is the left and right part of the system where blocks of even size centered around the middle of the chain have been removed. Color encodes the used interaction strength $J$ and markers encode the disorder strength $\Delta$. Each data point is an average over 100 realizations. Dotted lines are a guide to the eye. Solid lines show linear fits of the data for $\Delta = 0.1$, which allow to extract the localization length $\xi$.

(b) Logarithmic support $\log_{10}\mathrm{supp}(\mathcal{X}_{m-1}\mathcal{Z}_m\mathcal{X}_{m+1}, S)$ for $\eta = 1$ on $N = 11$ sites. , where $S$ is the middle of the system where sites to the left and right were successively removed. Color encodes the used interaction strength $J$ and markers encode the disorder strength $\Delta$. Each data point is an average over 100 realizations. Dotted lines are a guide to the eye.

Figure 4: Additional numerical data.

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
