# Peer review of "Edge mode locality in perturbed symmetry protected topological order"

_SciPost Physics, doi:SciPost Phys. 6, 072 (2019)_

## Round 1 · Referee Report · Anonymous · 2019-3-3

Strengths

1- interesting investigation of stabilization of topological order by disorder

Weaknesses

1- very hard to read, intricate notation
2- no finite size scaling
3- insufficient numerical quality to back claims

Report

The authors investigate a topological one dimensional quantum system in the presence of interactions and disorder. This system can in the absence of disorder be diagonalized exactly using a Jordan-Wigner transformation to Majorana operators and it is found in this limit that there exist edge modes due to the topological nature of the problem. The problem can still be diagonalized exactly in the presence of XX and YY interactions, but is promoted to a full quantum many-body problem in the presence of additional ZZ interactions.

The central problem investigated in this article is the locality of edge modes in the presence of interactions and disorder, and previous claims that disorder stabilizes topological order are challenged. For this, a method introduced in a previous paper by the authors to obtain local integrals of motion of the many-body localization problem is adapted to find conserved edge operators and their locality is studied by introducing a measure of locality, which compares the obtained operator to a truncation to a subset of the system.

The authors claim that the support of the edge modes acquires exponential tails into the bulk of the system, leading to no disorder dependence in the noninteracting case, but to stronger localization of some edge modes in the presence of disorder

In general, I find the paper interesting, since it seems to be a fresh approach to the problem of edge modes.
Unfortunately, the extensive use of mathematical notation and jargon makes it quite hard to read and I strongly suggest to explain the main ideas and the method in plain english before casting them into equations. Also, the notation should be explained this way, examples are the set complements, first featuring in Eq (16), as well as what stands behind Eq. (14), which is unclear to me. Why is there a floor function (?) appearing here?

Claims about exponential (in what?) corrections of the vanishing commutator of the edge mode operators with the Hamiltonian were made, and I would find it useful to see a numerical verification of this claim.

The description of the algorithm for the construction of the edge modes is really not clear and I think it should be extended considerably and explained in more detail, not relying on the previous work by the authors (one particularly unclear point is why the order of eigenvectors should matter).

In the final part of the paper, numerical calculations of edge mode operators where performed and the localization is checked by considering the supp(B,S) support quantifier. It is constructed such that it yields one if the operator is entirely supported on the subsystem S and is smaller than one otherwise. The decay of the support with decreasing subsystem size is used to quantify the decreasing operator weight at long distances. It would be useful to add an illustration of the subsystems used here.

The support does not decrease smoothly, there are considerable jumps and plateaus visible. Is it clear where they stem from?

The authors analyze the decay of the support in terms of exponential functions which are fitted to the numerical data. While in Fig. 1 this appears plausible (minus the jumps in the support function), I have a hard time believing this analysis in Fig 2 in the presence of interactions. Here, essentially three points are used to analyze the exponential decay, which are hardly sufficient to get a trustworthy result, in particular since they often don't even seem to represent a good fit.
I think what is dearly missing in this analysis is a careful study of the dependence of the results on system size. It is clear that there is considerable leakage of certain edge modes in the bulk of the system and it is therefore crucial to increase the size as much as possible, given the state of the art of full diagonalization, I believe that at least system sizes up to L=15 should be feasible for this 2^L Hilbert space. This would add at least another data point for the analysis of the exponential decay and make it more plausible. In addition, the stability with system size should be analyzed, comparing results for different chain lengths.

In Fig. 1 right there appears to be an artifact in the errorbars (?), which vanish suddenly at 10^-4. What is the reason for this?

In conclusion, I cannot recommend the publication of this paper in SciPost Physics in its current form for the following reasons: The introduced method for finding conserved operators is not new; the readability of the work could be improved; the numerical results are not convincing and should be extended in terms of system size and in particular by a careful finite size scaling analysis due to the severe limitations in system sizes. I suggest that the authors implement these recommendations in their changes to the manuscript before resubmission.

Requested changes

see report

---

## Round 1 · Referee Report · Anonymous · 2019-3-15

Strengths

1. Providing interesting results concerning the question how disorder, interactions and their mutual interplay affect localized edge modes

Weaknesses

1. While the results are obtained for a specific (class of) spin system(s) , the statements in the abstract are made in a possibly too general way.

Report

I read the manuscript with interest and think that, on the whole, it is an interesting contribution to the subfield of topologically-ordered systems and in particular to the question whether perturbations (through disorder and/or interactions) affect the localizing behavior of edge modes.
The authors show numerically for the specific example of a disordered XZX cluster hamiltonian (based on a spin chain) that an additional perturbation J tend to delocalize the edge modes, while the character of J turns out to be important: If J leaves the system non-interacting, this delocalization is essentially disorder independent, while the interacting counterpart shows an additional disorder dependence.

However, the conclusions drawn from the results based on their specific spin model are presented, both in the abstract and in parts of the text, in a rather general way. I think one should be a bit more careful with the level of generalization.

Also, in the abstract and several times in the text, it is stressed rather generally that "the narrative that disorder necessarily stabilizes topological order" is challenged. While this is the case in certain instances (see Ref. [16-18]) the way this opinion is put forward in this paper implies that this is the general believe for topological systems. On the other hand, too strong disorder in weakly gapped systems has also been shown to spoil topological behavour. As another example one could mention the physics of the topolgical Anderson insulator (see eg Groth et al, Phys Rev Lett 2009): While it also shows a disorder-induced transistion into a topological phase, the overall phase diagram is much more complicated.

Requested changes

1. Referring to both, the prefactor J in front of the perturbation and the parameter $\eta$, as an interaction is a bit unfortunate and might be confusing: For instance, in the caption to Fig. 2c) (last sentence in figure caption) the authors talk about "non-interacting results" (meaning $\eta=0$) presented as a function of interaction strength $J$ in Panel c.

2. I did not understand how the time-dependent quantity $B_0(t)$ enters, see Eq. (15), and what is its meaning. Also replacing the time average there by a basis expansion looks like invoking an ergodicity argument, e.g. equilibration. But is this justified within this study of localization effects?

3. In Fig. 1c the color coding is nearly not distinguishable in the symbols forming the straight line.

4. Fig. 3c): Color code given in upper right box does not coincide with the red and blue symbols used in the panel.

5. Fig. 2a,b): Where does the symmetry with respect to site number comes from that does not exist for $\eta=0$, see Fig. 1?

6. Page 6, second paragraph: "red dashed-dotted" should be "grey dashed dotted"?

7. First sentence in abstract: ".. with a .. modes"

---

## Round 2 · Referee Report · Anonymous · 2019-4-9

Report

This is my second report. While the authors addressed and satisfactory answered all items marked as requested changes in my previous report, they did not comment on the points raised in the major part of my first report: I wrote there:

"However, the conclusions drawn from the results based on their specific spin model are presented, both in the abstract and
in parts of the text, in a rather general way. I think one should be a bit more careful with the level of generalization.

Also, in the abstract and several times in the text, it is stressed rather generally that "the narrative that disorder
necessarily stabilizes topological order" is challenged. While this is the case in certain instances (see Ref. [16-18]) the
way this opinion is put forward in this paper implies that this is the general believe for topological systems. On the other
hand, too strong disorder in weakly gapped systems has also been shown to spoil topological behavour. As another example one
could mention the physics of the topolgical Anderson insulator (see eg Groth et al, Phys Rev Lett 2009): While it also shows
a disorder-induced transistion into a topological phase, the overall phase diagram is much more complicated."

I expected (and still expect) the authors to respond to this criticism and possibly to adapt the manuscript accordingly.

---

## Round 2 · Author Response

Dear Editor and Referees,

We would like to thank you all for the detailed and upright review of our article. We greatly appreciate the opportunity to resubmit our work improved through the thorough and encouraging feedback by the two referees. We used this opportunity to also change the layout.

In the following, we would like to give a detailed breakdown of the steps we have taken to address the feedback of the referees hopefully simplifying the review of the resubmitted version.

Yours sincerely,
Marcel Goihl, Christian Krumnow, Marek Gluza, Jens Eisert and Nicolas Tarantino

---

## Round 2 · List of Changes

"1. Referring to both, the prefactor J in front of the perturbation and the parameter η, as an interaction is a bit unfortunate and might be confusing: For instance, in the caption to
Fig. 2c) (last sentence in figure caption) the authors talk about ”non-interacting results”
(meaning η = 0) presented as a function of interaction strength J in Panel c."

In order to clarify this confusion, we extended the discussion of the Hamiltonian act-
ing as a perturbation introduced in Eq. (4). Furthermore, we added an additional dis-
claimer in the caption that the interaction strength may here also be referred to as
hopping strength.

"2. I did not understand how the time-dependent quantity B_0(t) enters, see Eq. (15), and what is its meaning. Also replacing the time average there by a basis expansion looks like invoking an ergodicity argument, e.g. equilibration. But is this justified within this study of localization effects?"

The operator given in Eq. (15) is the equilibrium representation of any B for non-
degenerate systems. Due to the topological degeneracies, we also require a rediago-
nalization of the subspaces in the basis of the constants of motion as explained below
Eq. (15). While this is basically only a definition, it is not a priori clear, that an operator
will equilibrate towards this form under time evolution. This is however expected for
interacting systems, even such that localize as we now also discuss below Eq. (15) and
cite the corresponding result. These operators serve as an ansatz for our algorithm
only and even work in a regime, where equilibration is not expected (non-interacting,
disordered systems).

"3. In Fig. 1c the color coding is nearly not distinguishable in the symbols forming the
straight line."

We have added a comment on this in the caption.

"4. Fig. 3c): Color code given in upper right box does not coincide with the red and blue
symbols used in the panel."

Here, the red symbols are actually on top of the orange ones, which we now also
indicate in the caption.

"5. Fig. 2a,b): Where does the symmetry with respect to site number comes from that does not exist for η = 0, see Fig. 1?"

We thank the referee for pointing this out. The symmetry is inherited from the sym-
metry of the set S we use to calculate the support. Since for the interacting system,
we always need to construct products of edge modes, the support is predominantly
on both edges. This is why we cut out boxes centered around the middle of the chain
to calculate the support which in turn causes the symmetry in our plots. For compar-
ison: In the non-interacting case, we work with the probably more intuitive shape of
S which is just a consecutive region starting at one edge. We have added an explanation towards the end of section 3.3 and a discussion in the results section (4.2) of the interacting model. Further comments on typos were fixed.

Anonymous Report 2 on 2019-3-3

"Unfortunately, the extensive use of mathematical notation and jargon makes it quite
hard to read and I strongly suggest to explain the main ideas and the method in plain
english before casting them into equations. Also, the notation should be explained this
way, examples are the set complements, first featuring in Eq (16), as well as what stands
behind Eq. (14), which is unclear to me. Why is there a floor function (?) appearing here?"

We thank the referee for this feedback and overhauled the presentation of mathemat-
ical details according to it. As this was also in similar fashion pointed out by the other
referee, we added an explanation of the sets used for the support calculation in section
3.3. Moreover, we put a more colloquial explanation below Eq. (14).

"Claims about exponential (in what?) corrections of the vanishing commutator of the
edge mode operators with the Hamiltonian were made, and I would find it useful to see
a numerical verification of this claim."

Indeed, we missed to write ”exponential in system size” at one point in the document
and are thankful for the referee to spot this error. The failure of the edge modes to
commute with the Hamiltonian is in fact guaranteed by Kramer’s theorem. A clarification as to why this is the case has been added to Section 2. Investigating this numerically in the η = 1 case is not possible, as we would need to recover the individual edge modes, and not the bilinears which are available to us.

"The description of the algorithm for the construction of the edge modes is really not clear and I think it should be extended considerably and explained in more detail, not relying on the previous work by the authors (one particularly unclear point is why the order of eigenvectors should matter)."

In order to clarify this part, we have moved all reference to the MBL construction
towards the end of section 3.2 and added more detail to the main steps of the algorithm.
Specifically, we now point out that since any constant of motion can be mapped onto
the Pauli-Z operators only the order of the eigenvectors determine its locality.

"The decay of the support with decreasing subsystem size is used to quantify the decreasing operator weight at long distances. It would be useful to add an illustration of the subsystems used here."

This points is similar to the first one, which we hope to sufficiently incorporate by
putting in additional description of the sets.

"The support does not decrease smoothly, there are considerable jumps and plateaus visible. Is it clear where they stem from?"

While we previously ascribed these plateaus intuitively to the three-locality of the
Hamiltonian, this remark motivated us to actually show its origin for the non-interacting perturbation in the infinite systems limit. The calculation can be found in an additional appendix and we furthermore added explanatory sentences into the main text.

"I think what is dearly missing in this analysis is a careful study of the dependence of the results on system size. It is clear that there is considerable leakage of certain edge modes in the bulk of the system and it is therefore crucial to increase the size as much as possible, given the state of the art of full diagonalization, I believe that at least system sizes up to L=15 should be feasible for this 2^L Hilbert space. This would add at least another data point for the analysis of the exponential decay and make it more plausible. In addition, the stability with system size should be analyzed, comparing results for different chain lengths."

We agree with the referee that in the previous version, we missed out on providing
standard evidence for the reliability of our algorithm. In the present resubmission
we added system size scaling for the non-interacting code. Due to plateau structure
that also persists in the interacting case we essentially only obtain very few points
for fitting the exponential envelope. Making the system size smaller will reduce this
to only two points, which results in very unstable data. Enlarging system size on the
other hand is not feasible either because the bottle neck is unfortunately not the exact
diagonalization but rather the rediagonalization of the 2^(L−2) many 4 × 4 matrices. A
single realization takes 8 days to construct all three edge modes on 12 sites. We have
added an explanation of this problem to the resubmission as well.

"In Fig. 1 right there appears to be an artifact in the error bars (?), which vanish suddenly at 10^−4 . What is the reason for this?"

We are thankful for this remark as we missed putting an explanation into the first
submission. The error bars are caused by a very sharp dropoff of the support due to
the small interactions which lead to a less stable fit. We added such an explanation in
the result description in section 3.1 as well.

---

## Round 3 · Referee Report · Anonymous (Referee 2) · 2019-5-19

Report

The authors significantly expanded the text and I find the discussion of the algorithm now much more clear and accessible.
They also added a discussion of the decay of the operator support, explaining the plateaus visible in the numerical data. The discussion of the numerical procedure, system sizes and statistical analysis was improved.

In summary, I find that the referee comments have been addressed in
a satisfactory manner and I recommend the article for publication in SciPost Physics.

---

## Round 3 · Referee Report · Anonymous (Referee 3) · 2019-6-4

Strengths

I think that after the further revisions of the manuscript, and in view of the convincing answer of the authors, the paper now deserves publication. Hence i recommend to accept the paper.

Report

I recommend publication.

---

## Round 3 · Author Response

Dear readers,

we have included the latest feedback of the referee and included a discussion of our fitting errors.
As pointed out in our comment of version 2, we caused some misunderstanding by using the term 'stability'. We have now included a more detailed discussion on that subtle issue and went on describing the phenomenon we want to study as 'localization' since 'stability' is common to describe phases. Moreover it occurred to us, that we did not explain the error bars we obtain from our fitting procedure. These are least-squares errors of the fitting algorithm and hence strengthen our claim that the data for the interacting system is meaningful, even though the amount of available data points is rather small.

We look forward to hearing from you,
the authors

---

## Round 3 · List of Changes

• added a discussion in the introduction on why localization of the edge modes is crucial for employing these systems for quantum information tasks.
  • removed claims about a 'challenged narrative' altogether.
  • added a discussion of the fitting errors in the captions of the plots as well as in the result section.

---

## Editorial Decision

published